# A Protocol for Comprehensive Analysis of Gait in Individuals with Incomplete Spinal Cord Injury

**DOI:** 10.3390/mps7030039

**Published:** 2024-05-04

**Authors:** Emelie Butler Forslund, Minh Tat Nhat Truong, Ruoli Wang, Åke Seiger, Elena M. Gutierrez-Farewik

**Affiliations:** 1Department of Neurobiology, Care Science and Society, Karolinska Institutet, 141 83 Stockholm, Sweden; emelie.forslund@aleris.se (E.B.F.); ake.seiger@ki.se (Å.S.); 2Aleris Rehab Station R&D Unit, 169 89 Solna, Sweden; 3KTH MoveAbility, Department of Engineering Mechanics, KTH Royal Institute of Technology, 100 44 Stockholm, Sweden; minht@kth.se (M.T.N.T.); ruoli@kth.se (R.W.); 4Department of Women’s and Children’s Health, Karolinska Institutet, 171 77 Stockholm, Sweden

**Keywords:** paraplegia, gait, ambulation, movement analysis, machine learning, EMG, predictive modeling

## Abstract

This is a protocol for comprehensive analysis of gait and affecting factors in individuals with incomplete paraplegia due to spinal cord injury (SCI). A SCI is a devastating event affecting both sensory and motor functions. Due to better care, the SCI population is changing, with a greater proportion retaining impaired ambulatory function. Optimizing ambulatory function after SCI remains challenging. To investigate factors influencing optimal ambulation, a multi-professional research project was grounded with expertise from clinical rehabilitation, neurophysiology, and biomechanical engineering from Karolinska Institutet, the Spinalis Unit at Aleris Rehab Station (Sweden’s largest center for specialized neurorehabilitation), and the Promobilia MoveAbility Lab at KTH Royal Institute of Technology. Ambulatory adults with paraplegia will be consecutively invited to participate. Muscle strength, sensitivity, and spasticity will be assessed, and energy expenditure, 3D movements, and muscle function (EMG) during gait and submaximal contractions will be analyzed. Innovative computational modeling and data-driven analyses will be performed, including the identification of clusters of similar movement patterns among the heterogeneous population and analyses that study the link between complex sensorimotor function and movement performance. These results may help optimize ambulatory function for persons with SCI and decrease the risk of secondary conditions during gait with a life-long perspective.

## 1. Introduction

A spinal cord injury (SCI) is a life-changing event for both the individual and his/her family. A traumatic SCI, defined as “any injury to the spinal cord that is caused by trauma or damage resulting from an external force” [1], can occur due to events such as falls, traffic, or sports accidents, while non-traumatic SCI can be caused by, e.g., myelitis, infections or vascular events [2]. The SCI can be either complete or incomplete, with a highly variable level of remaining function below the level of injury. In general, individuals with SCI have complicated disabilities that require multidisciplinary efforts and life-long follow-up. 

SCI is associated with impaired muscle function and lack of sensation at and below the level of injury, and decreased or absent ambulatory function is one of the most characteristic consequences. However, there are vast individual differences ranging from requiring a manual or electric wheelchair to the ability to walk with or without assistive devices.

The level and extent of spinal involvement determine the ability to stand and walk, wherein muscle function in knee extensors, corresponding to a neurological level of lumbar 3, is regarded as crucial for functional ambulation [3,4]. Particularly with incomplete SCI, motor and sensory function can differ substantially between individuals, making prognosis regarding ambulatory function complicated. Around 80% of persons with incomplete SCI (American Spinal Injury Association Impairment Scale D) reach a level of independent walking [5]. 

Walking after SCI with impaired motor and/or sensory function, often in combination with spasticity and pain, is likely to require a greater effort with higher energy consumption than normal [6], thus limiting walking capacity and performance. A slower walking speed has been reported in persons with incomplete SCI [6,7,8] and has been interpreted as a means to maintain balance during walking [8]. Occurrence and fear of falling are both common [9,10], and slower walking speed has been associated with an increased risk of falling in both persons with SCI [9] and others [10]. Mode of mobility can also be influenced by both intrinsic factors, such as BMI and age, as well as external factors, such as lack of adaptations in the environment and access to technical aids. There is a complex interplay between physical prerequisites and environment, but psychological factors such as motivation and fear of falling can also play a role in an individual’s choice of mode of mobility [3]. Further, it is not uncommon for a person with impaired muscle function in the lower extremities due to SCI to use different technical aids and orthoses to perform their daily activities, depending on the task. Various robotics with different levels of support can also be used for gait training and in laboratory settings but are rarely used in home environments. Whether a patient primarily uses a wheelchair or walks, optimizing their individual ability for functional independence, health, and participation with a life-long perspective should be the main focus of rehabilitation interventions [1]. 

As a result of medical achievements, better rehabilitation, and life-long follow-up, persons with SCI live longer after their injury, and the prevalence is thus increasing. The age at injury has increased during recent years [11], leading to challenges during rehabilitation due to previous injuries and diseases with a lower expected outcome [5]. There is also an increased frequency of these incomplete injuries with various levels of impaired function in the lower extremities [12]. Altogether, this leads to an increasing number of persons with spinal cord injury with remaining but impaired gait function. The need for continuous rehabilitation and exercise is therefore crucial, and existing clinical guidelines must be constantly updated and should also be integrated with new technologies.

Ambulatory function in persons with incomplete SCI is heterogeneous with varying extent of remaining motor and sensory function, complicated by, e.g., spasticity. It is a challenge to optimize the rehabilitation. In addition to clinical evaluations by experienced rehabilitation staff, detailed objective measures of different aspects of gait are also necessary to facilitate further improvements of clinical interventions, walking aids, and orthoses to ultimately improve ambulatory function in persons with SCI. Instrumented 3D movement analysis in combination with electromyography (EMG) and measurement of ground reaction forces enables detailed studies of muscle and locomotor function during gait. In order to better understand movement strategies and the consequences of SCI, mathematical models that classify and characterize gait patterns and walking economy can be developed and may, in a long-term perspective, refine gait training and the use of technical aids and orthoses. A clearer picture of these factors and how they contribute to different aspects of gait could help the clinical staff to tailor their rehabilitation interventions, e.g., to primarily focus on improving muscle strength or reducing spasticity.

Data-driven analyses such as machine learning are more and more frequently applied in clinical settings driven by the exponential growth of electronic health data and computational power. For the specific population of persons with SCI, machine learning algorithms have demonstrated their potential in diagnosis, prognosis, and discovery of subgroups. In a recent systematic review [13], 39 studies that apply machine learning algorithms in this population were identified, wherein 34 used mostly convolutional neural networks on imaging data with a focus on diagnosis, and 5 employed tree ensemble algorithms trained on clinical tabular data to address prognosis. Unsupervised learning via clustering analysis can also help clinicians identify subgroups among patients with SCI. Basiratzadeh et al. [14] identified five subgroups from 334 patients with traumatic SCI using spectral clustering on their baseline variables. Werner et al. [15] identified four subgroups from 66 patients with SCI using k-means clustering on sensor-derived gait features. These studies demonstrate the potential of machine learning algorithms to enhance our understanding beyond traditional clinical assessments.

Regaining and optimizing ambulatory function after SCI has been a major goal in both rehabilitation and research over the years, but as both the prevalence of persons with incomplete SCI [12] and age at injury [11] are increasing, the physical prerequisites for regaining and refining ambulatory function is also changing. There has been a focus on advanced equipment such as body weight-supported treadmill training, robotics, and exoskeletons in recent research. In addition to their direct use in rehabilitation, robotics can also be useful in rehabilitation by assessing stiffness, joint torque, and range of motion in patient populations, metrics that are not trivial to assess; a recent report describes their use in a stroke population [16]. Despite the advantages of providing an opportunity for massive and repetitive gait training, the high cost and need for staff with high technical skills [17] make their frequent use impractical on a large scale outside the rehabilitation centers. It thus remains important to perform continuous research regarding overground ambulation for persons who walk in daily life. Moreover, there are rapid developments in technology to measure and analyze movements, muscle function, and forces during ambulation. We believe that leveraging the use of new technologies with the change in patient characteristics could bridge the gap of knowledge in this field. 

Transdisciplinary collaboration is necessary to accomplish this task and is proposed in the current protocol of a joint venture between the Promobilia MoveAbility Lab at the KTH Royal Institute of Technology, The Spinalis Clinic, part of the Aleris Rehab Station, and Karolinska Institutet, all in Stockholm, Sweden.

With this project, we aim to improve the basis for rehabilitation by contributing to a better understanding of the options for regaining motor function and how to maximize ambulatory performance by integrating new technology with rehabilitation and training. The long-term goal is to enhance clinical guidelines regarding ambulation with and without assistive gait devices and orthoses and to develop targeted interventions in persons with incomplete SCI—a complex, heterogeneous, and growing group. This article describes the investigative design of the project.

Specific study questions include the following:-Do correlations exist between various physical characteristics such as muscle strength, spasticity, walking speed, and energy economy?-Are there relationships between various physical characteristics and movement strategies and performance?-Can data-driven machine learning algorithms identify similarities and patterns in the gait patterns in this study population?-Can data-driven machine learning algorithms identify complex relationships between parameters beyond those identified by standard statistical analyses?

## 2. Experimental Design

### 2.1. Setting/Team

A multidisciplinary project group was formed, including scientists with backgrounds in biomechanical engineering, neurophysiology, and clinical rehabilitation, for an approach that encompasses measurements of different aspects of gait, including 3D movement analysis, EMG, muscle strength, and energy expenditure. This study is designed as a cross-sectional observational cohort study.

The partners in this protocol contribute with complementary research expertise crucial to the study goals. The Spinalis Clinic, part of Aleris Rehab Station, is a unique rehabilitation center for persons with SCI, starting from primary rehabilitation to lifelong follow-up. Aleris Rehab Station is the largest neurological rehabilitation center in Sweden, and the Research and Development Unit at Aleris Rehab Station is part of the Karolinska Institutet, which has highly specialized expertise in neurological diagnoses. Promobilia MoveAbility at the KTH Royal Institute of Technology is an internationally recognized biomechanics research group focusing on decoding, analyzing, and enhancing human movement. The group’s research focuses on people with various neurological diagnoses, including developing assistive robotic devices, measuring, modeling, predicting movement, and characterizing motor disabilities such as spasticity.

The proposed protocol involves the following areas of investigation (Figure 1) in individuals with incomplete SCI with a wide range of motor deficits:-Clinical assessment of neurological and motor function, including injury-level classification and measurement of sensory and motor function;-Energy expenditure and heart rate variability during 6 min walk test (6MWT) [18] and evaluating whether associations exist to physical variables such as muscle strength in lower extremities;-Computerized 3D movement analysis during gait at self-selected speed, with and without habitual walking aids and/or orthoses;-Characterization and classification of gait with data-driven approaches;-Joint torque and motor unit behavior analysis in the lower limbs.

### 2.2. Participants

We aim to recruit 40 adults with incomplete paraplegia due to SCI, with neurological deficits in the lower extremities. Included participants should be able to walk independently at least 30 m with or without assistive devices and/or orthoses and should walk habitually in their daily lives. All participants will be enrolled at the Spinalis Clinic at Aleris Rehab Station (Rehab Station). Inclusion criteria are 18–75 years of age, acquired SCI at least one year prior or shorter if considered as having a neurologically stable condition, and ability to participate in the extensive study protocol.

Exclusion criteria will be presence of other conditions affecting gait, no neurological sequels after their SCI, and persons who do not walk in their daily life or who walk merely during training sessions.

Additionally, a convenience sample of persons without SCI with matching age and gender distribution will be recruited to provide reference material.

### 2.3. Sample Size

To enhance ecological validity, the recruitment procedure will be performed in conjunction with regular clinical procedures of regular SCI follow-up programs at Rehab Station. The sample size of 40 was chosen for several reasons. In lieu of pilot data to base statistical power analyses on, the research team and the clinical staff at the SCI unit estimate that around 40 persons will be required to capture the broad spectrum of gait patterns. Furthermore, this number agrees well with similar studies within the field [15,19,20,21].

### 2.4. Ethics

Ethical permission has been obtained from the Swedish Ethical Review Authority (Dnr. 2020-02311, 2020-07067, and 2022-00629-02). All participants will sign an informed consent after receiving oral and written information. The Helsinki Declaration will be followed.

## 3. Procedure

The study participants will be consecutively invited to participate in the study during regular visits to the Spinalis Clinic. After signing an informed consent, the first visit will take place at the Spinalis Clinic; see Figure 2 for overview of data collection. SCI level and extent will be classified according to international standards for the classification of SCI and the American Spinal Injury Association impairment scale [22,23] by an experienced physiotherapist (>25 yrs of SCI rehabilitation). Muscle strength (manual muscle test on a 0–5 scale [24]) and spasticity [25,26] will be evaluated, and 6MWT [20] with simultaneous registration of oxygen consumption will be performed.

The second visit will take place at the Promobilia MoveAbility Lab, KTH Royal Institute of Technology. Muscle activation in major lower extremity muscles will be measured with EMG, and strength will be simultaneously measured with a handheld dynamometer during maximal voluntary isometric contractions in several positions and during several functional tests. In addition to EMG recording, whole-body motion analysis will be performed, and ground reaction forces will be measured. Each participant will be asked to walk several times along a 10 m walkway, to stand, and to perform various functional tasks such as the five-time sit-to-stand test [27]. Finally, the activity of the dorsiflexor and plantarflexor muscles will be recorded during maximal and submaximal isometric ankle contractions with high-density EMG grids. All methods are described in greater detail in the next section. The participants will be offered intermittent breaks during the data collection as needed.

All methods and measurement systems are well established and have been used clinically and scientifically in persons with different neurological diagnoses. The study protocol will be thoroughly piloted by the research team for this population.

### 3.1. Measurements and Assessments of Parameters at Rehab Station

In order to describe the participants’ SCI adequately, international standards for neurological classification of spinal cord injury (ISNCSCI) [22,23] will be used for standardized neurologic assessment of the motor and sensory deficit after SCI, and the extent of injury is classified according to American Spinal Injury Association (ASIA) Score (AIS) as A-E. Motor function has been shown to be closely correlated to ambulation [7]. Lower extremity motor score (LEMS) of this ISNCSCI classification, i.e., voluntary muscle strength of the five key muscles groups (hip flexors, knee extensors, ankle dorsiflexors, toe extensors, ankle plantarflexors) of both lower extremities, will be graded on a 0–5 scale for a maximum score of 0–50 [22,23]. Proprioception of hips, knees, feet, and first toes will be manually tested and classified as normal, impaired, or absent according to the recommendations of the data sets from ISNCSCI.As manual muscle testing is common in clinical routines, muscle strength in the legs will be assessed with the commonly used Daniels and Worthingham [24] scale. Also, the five-time sit-to-stand test [27] will be used. The test has been previously used for SCI research and has shown good reliability and validity [28,29].As spasticity is a common problem during walking after SCI, spasticity in the lower extremities will be assessed by the Modified Ashworth Scale on a 0–5 scale [25,26].In order to systematically classify the use of walking aids and orthoses, the Walking Index for Spinal Cord Injury WISCI-II (maximal ability of a person to walk 10 m) will be used [30]. Intra-rater and inter-rater reliability have been considered excellent at 1.0 and 0.98, respectively [31], and a change of one level has been considered clinically significant [32].According to Perry [33], a functional ambulation requires the ability to walk 300 m in daily activities at a speed of 40–100 m/min. Global walking endurance will be assessed with the 6MWT, which has been reported as valid and reliable for persons with SCI [18,34]. The participants will walk 30 m laps in an indoor corridor on a hard, flat surface for 6 min, and the total distance walked will be recorded. Recommended standardized instructions and environment will be used.As ambulatory persons with SCI are expected to have a higher energy cost (lower walking energy economy) [6], breath-by-breath oxygen and carbon dioxide ratios will be measured during the 6MWT with a spiroergometric device (Metamax 3B-r2, Cortex). The values during resting 10 min prior to and following the test will also be recorded. The values during the last two minutes will then be averaged. The subject will also wear a chest strap heart rate sensor (Polar H10, Polar Electro, Bromma, Sweden) to monitor heart rates during the process. The Borg CR scale 0–10 [35] will be used for subjective grading of exertion before and after the 6MWT.

### 3.2. Assessments of Parameters at Promobilia MoveAbility Lab

To complement manual muscle testing with a more objective assessment, muscle strength in the legs will be measured with a handheld dynamometer (MicroFET2, Hoggan Scientific, Salt Lake City, UT, USA). The following muscles will be measured: hip abductors, hip flexors, hip extensors, knee flexors, knee extensors, ankle dorsiflexors, and ankle plantarflexors during maximal voluntary isometric contraction trials. Corresponding moment arms will be measured to compute the maximal exerted torques.Detailed information on muscle activity during walking will be measured with surface EMG. Muscle activity during maximal voluntary isometric contractions and during movement analysis will be measured with surface differential EMG transmitters (myon/Cometa aktos nano, Bareggio, Italy) placed over ten major lower extremity muscles per side with sensor placement according to the SENIAM guidelines [36]. These include gluteus maximus, gluteus medius, rectus femoris, vastus lateralis, biceps femoris, tibialis anterior, gastrocnemius medialis, gastrocnemius lateralis, soleus, and peroneus longus.To explore the different movement patterns of persons with incomplete SCI, 3D marker trajectories will be collected with a 10-camera system with 100 Hz sampling frequency (Vicon V16, Oxford, UK) with whole-body placement according to the CGM2 model [37], with 55 markers for calibration and 49 markers for tracking movement). Ground reaction forces will be measured with floor-mounted force plates (AMTI Optima, Boston, MA, USA), from which joint kinetics will be computed via inverse dynamics during walking and balance tasks. Participants will be asked to walk at their preferred speed with their habitual daily assistance devices and, when possible, with no or as few assistive devices as possible. Dynamic standing will be evaluated while the participant stands on two force plates.To study patterns of lower leg muscle activation in detail over a large area, a high-density 64-channel surface arrays (HD-EMG, Quattrocento, OT Bioelettronica, Turin, Italy) will be used over the ankle dorsiflexors and ankle plantarflexors in a seated position during submaximal isometric contractions with visual biofeedback via a computerized serious game.

### 3.3. Data Management and Analysis

During clinical assessments of muscle strength, with respect to the classification of the SCI, the lower extremity sum score from the international standards for neurological classification of spinal cord injury (ISNCSCI) will be used [38]. A sum score of individual manual muscle test scores for each leg will be computed, as previously described [18,39]. Kim et al. [39] have recommended the use of a sum score for the stronger leg, but we will compute the sum score for both legs. To evaluate sensitivity, the S1 dermatomes from the ISNCSCI (normal, impaired, or absent) representing the sole of the feet will be computed with a sum score for both feet. To evaluate proprioception, a sum score for hip, knee, feet, and big toes will be computed [40]. To evaluate spasticity, a sum score of the modified Ashworth scale for both legs will be computed, as reported previously [41].

Movement parameters and performance will be computed via temporospatial parameters, oxygen consumption, and gait patterns, for example, with the Gait Deviation Index [42]. Gait patterns will also be classified and characterized using data-driven machine learning techniques, as described in the next section. 

### 3.4. Statistics

Background data will be described according to descriptive and inferential statistical analyses, with parametric or non-parametric statistics with subsequent post hoc corrections for small samples, as deemed most relevant for the sample size and after evaluating the data for normal distribution by Shapiro–Wilks test as well as visual inspection. Interval scale variables with a distribution close to normal will be presented with mean and deviation. Non-normally distributed interval scale variables will be presented with median and lower and upper quartiles (25th and 75th percentile). Differences between subgroups will be analyzed using an independent sample *t*-test or Mann–Whitney U-test for non-parametric data.

Potential associations between physical assessment parameters such as muscle strength, sensitivity, proprioception spasticity, and movement outcome parameters such as walking speed and energy economy will be computed with statistical correlation tests, specifically Spearman (non-parametric) and multiple regression analyses, based on the data distribution.

### 3.5. Machine Learning Algorithms

To advance clinical understanding of SCI through machine learning techniques, we plan to apply the techniques across three potential applications: pattern discovery via clustering analysis, diagnosis with classification algorithms, and outcome prediction with regression algorithms.

For clustering SCI subgroups, data mining techniques such as k-means and hierarchical clustering can be considered [43,44]. It is of interest to characterize each cluster’s unique attributes. As clustering may violate normality assumptions, non-parametric Kruskal–Wallis (continuous variables) and chi-squared (categorical) tests can be considered to determine the significance between clusters. When overall significances are found in clinically relevant variables, post hoc comparisons will be used to further determine pairwise cluster differences.

For classification and regression applications, the literature has shown that tree-ensemble models are effective on tabular data [45]. Therefore, we plan to investigate several tree-ensemble models, such as random forests [46] and XGBoost [47], for these applications and train them through our collected data sets. Several metrics will be used to evaluate model performance and select the best for each application. For classification, evaluation metrics such as accuracy score, precision score, and recall score will be considered. Accuracy reflects overall correct predictions, while precision finds models minimizing Type I errors. Recall identifies models with the fewest Type II errors. For regression, root mean square error can be used to quantify average errors between predicted and actual values. R-squared determines the quality of fitness by explaining the variation. In both applications, we will perform grid searches for fine-tuning hyperparameters while training. For example, key hyperparameters for random forests, such as the number of trees, maximum tree depth, and measures of split quality, can be investigated. By exploring this hyperparameter space, the optimal configuration for each model can be achieved. Cross-validation will then be used to compare algorithms across relevant metrics. The classifiers or regressors averaging the highest scores will be determined as the most appropriate models for each application.

## 4. Expected Results

Energy expenditure and heart rate variability will be described during the 6MWT, and associations to physical variables such as muscle strength, sensitivity, proprioception, and spasticity will be explored. These results might help clinicians decide whether to recommend compensatory strategies such as orthoses or walking aids or to focus on modifiable factors such as muscle strength, reduction of spasticity, and/or fitness. Furthermore, it can be a foundation for future development of smart orthoses.

Movements and kinetics during gait at self-selected speeds will be analyzed and described in relation to muscle strength and SCI sensory and motor levels.

Detailed motor unit behavior during gait and during submaximal contractions will be described. These results will show how much of the maximal capacity is used when walking, which might help clinicians develop better long-term strategies for persons with incomplete SCI to optimize and maintain their ambulatory performance.

The heterogeneity of gait function will be characterized, which might help the clinicians to better understand the complexity and different needs of this group.

Associations between gait performance, oxygen economy, and underlying sensory and motor function will be identified to provide key information that is otherwise not immediately apparent from clinical evaluation.

### Significance

For persons with reduced muscle function, there is an increased demand for remaining muscles to walk, potentially leading to a risk for overuse injuries and pain. Also, in conjunction with the natural aging process, there is a risk of decline over time. To achieve and maintain as high a level of function as possible, all available measures should be taken in order to better understand the mechanisms regarding the complex interplay between muscle function, other factors, and ambulatory performance. The development of new technologies, such as active orthoses, assistive gait devices and robotics, requires in-depth knowledge of the gait mechanisms of persons with SCI with different types of muscle function.

The different parts of this project aim to explore and describe the interaction of various factors on different aspects of gait. With this inclusion criterion of ambulatory adults with incomplete SCI, we intend to describe and explore the vast interindividual variation and walking performance, from those relying heavily on walkers and orthoses to those with near-to-normal function. By inviting persons of different ages and ambulatory performance during their regular SCI follow-up program, we strive to enhance the representativity of the sample.

This thorough data collection will result in a large amount of complex data. Machine learning models are expected to complement and perform analyses beyond the traditional analyses and facilitate further exploration. As there are relatively few studies on walking in this population, we expect to contribute to the body of knowledge. Quantitative analysis of the complex relationships between sensory and motor disability and gait performance can greatly complement standard clinical practice.

This project can contribute to increased knowledge and help to form a base for future clinical guidelines that target appropriate measures and exercise programs and/or for designing assistive devices for persons with various degrees of impaired gait. Due to medical achievements, better rehabilitation, and life-long follow-up programs, persons with SCI live longer, and the SCI population is increasing. Furthermore, a growing proportion of SCIs occur at an older age and result in incomplete injuries and with some remaining walking function. Therefore, there is a high need to focus on issues that could improve ambulatory performance from a long-term perspective in order to enhance participation, improve quality of life, and maximize everyday functioning. These results have the potential to improve all of these goals so that persons with incomplete SCI can be as active as possible for as long as possible while trying to avoid secondary complications.

## Figures and Tables

**Figure 1 mps-07-00039-f001:**
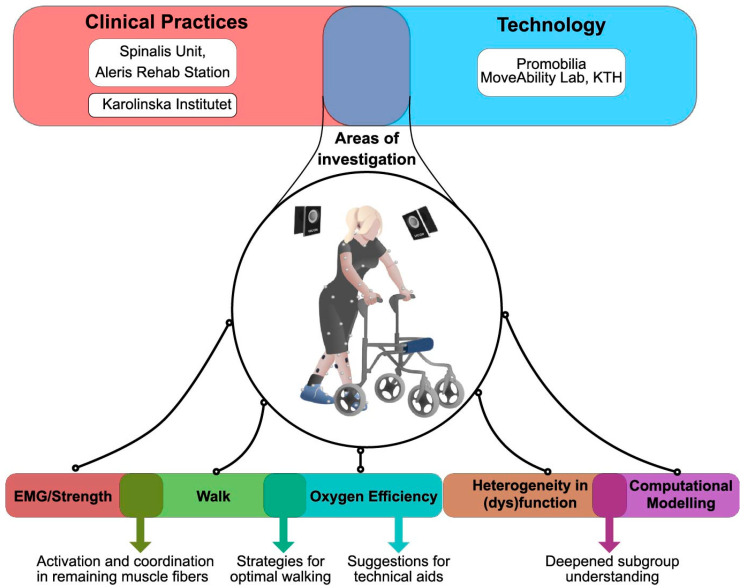
Overview of areas of interest for the project.

**Figure 2 mps-07-00039-f002:**
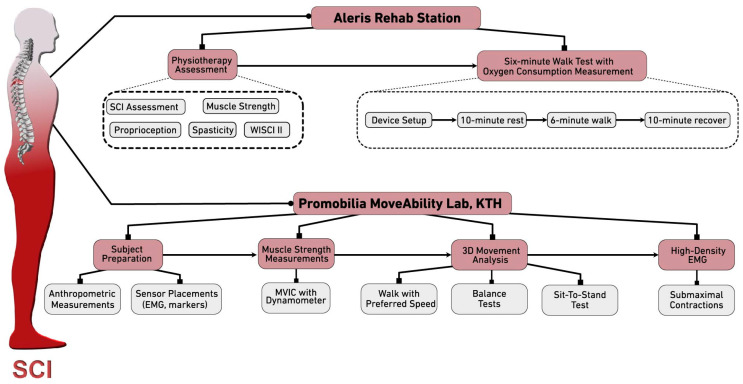
Overview of data collection at Rehab Station and Promobilia MoveAbility Lab.

## Data Availability

Not applicable.

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
