# Peer review of "A Protocol for Comprehensive Analysis of Gait in Individuals with Incomplete Spinal Cord Injury"

_mps, 2024, doi:10.3390/mps7030039_

Round 1

Reviewer 1 Report

Comments and Suggestions for Authors

I found the article "A protocol for comprehensive analysis of gait in individuals with incomplete spinal cord injury" interesting and unique in some way, but I feel very ambivalent about it. At first glance, this is just a simple detailed analysis plan in the context of a research project. And since I don't find, for me, interesting, theoretical justification in many choices or prognostic insights - I have doubts - about whether I would trust that plan or not. There are always uncertainties in research that can change the design: parameters and trends in individual subjects, and consistency and quality of data. In addition, I did not find any particularly innovative elements. Nevertheless, I can see the specifics of the protocol that may help other researchers to look at the situation in a broader or completely different way than the usual case. Therefore, I have a few comments and observations that I recommend the authors take into account.

1)      If the article is about a specific gait protocol, the authors must enter keywords that reflect this. I think gait or gait protocol must be there.

2)      I really miss the justification in this protocol everywhere. For example, if it is planned to invite 40 patients, then why this number; Was the sample size calculated or so? Because it will influence expected results a lot. If tools and equipment are chosen, then why are they needed and so. After all, a justification is possible with published individual works and achievements. I think, that a theoretical background is needed.

3)      It would be much more informative to present some kind of workflow or algorithm-type scheme for both options: in Rehab Station and Promobilia MoveAbility Lab.

4)      The data analysis part is very generalized. after all, all specified parameters can be named, which ones will require pre-processing, which ones will not, and if statistics will be used, then what parametric and non-parametric methods are specified? Quantitative parameters can be grouped, and their analysis methods can be specified.

5)      Expected results are also not specific. If it is expected to define the gait function, what specific research indicators will show it? It is necessary to link methods, data analysis and results with each other.

6)      One of the results of such a protocol should be conclusions that would indicate whether the protocol will be successful compared to the currently used research and how much more information it can provide. You also need to justify why you think so. Because sometimes more research doesn't mean more information. Sometimes you get the same information from the other side or in a different context.

Author Response

Thank you for thoughtful feedback. We have revised according to your comments and hope you agree that the manuscript has improved.

1) I found the article "A protocol for comprehensive analysis of gait in individuals with incomplete spinal cord injury" interesting and unique in some way, but I feel very ambivalent about it. At first glance, this is just a simple detailed analysis plan in the context of a research project. And since I don't find, for me, interesting, theoretical justification in many choices or prognostic insights - I have doubts - about whether I would trust that plan or not. There are always uncertainties in research that can change the design: parameters and trends in individual subjects, and consistency and quality of data. In addition, I did not find any particularly innovative elements. Nevertheless, I can see the specifics of the protocol that may help other researchers to look at the situation in a broader or completely different way than the usual case. Therefore, I have a few comments and observations that I recommend the authors to take into account.

If the article is about a specific gait protocol, the authors must enter keywords that reflect this. I think gait or gait protocol must be there.

Response: We have added gait as a keyword.

2) I really miss the justification in this protocol everywhere. For example, if it is planned to invite 40 patients, then why this number; Was the sample size calculated or so? Because it will influence expected results a lot. If tools and equipment are chosen, then why are they needed and so. After all, a justification is possible with published individual works and achievements. I think that a theoretical background is needed.

Response: We have added justifications and approximately 20 more references to improve and clarify the theoretical background throughout the manuscript. The number 40 was chosen for a few reasons. It agrees with participant count in similar studies. We have also performed statistical power analyses for gait studies in other subject populations, and frequently determine an acceptable subject count of around 30-40; we did not write this specifically in the protocol article, since this we would need pilot data and very specific study hypotheses to determine the required participant count. And last, for practical reasons; we know from experience that it will probably take a few years to include this many participants.  We have added a section to motivate the sample size.

3) It would be much more informative to present some kind of workflow or algorithm-type scheme for both options: in Rehab Station and Promobilia MoveAbility Lab.

Response: We have added an overview of the workflow during data collection, see Figure 2.

4) The data analysis part is very generalized. after all, all specified parameters can be named, which ones will require pre-processing, which ones will not, and if statistics will be used, then what parametric and non-parametric methods are specified? Quantitative parameters can be grouped, and their analysis methods can be specified.

Response: The revised manuscript contains a more detailed description of the planned data analysis. Please see section 3.

5) Expected results are also not specific. If it is expected to define the gait function, what specific research indicators will show it? It is necessary to link methods, data analysis and results with each other.

Response: We have added a more detailed description in the section of preliminary results and tried to clarify the link to the methods and analysis sections by using the same structure. Please see methods, analysis and results sections.

6) One of the results of such a protocol should be conclusions that would indicate whether the protocol will be successful compared to the currently used research and how much more information it can provide. You also need to justify why you think so. Because sometimes more research doesn't mean more information. Sometimes you get the same information from the other side or in a different context.

Response: Interesting point. Of course we agree in general. This, however, is a field where technologies are developing rapidly measurement systems, analysis approaches and data-driven models. The body of research regarding ambulatory function in persons with incomplete SCI must likewise be constantly updated, as this population is changing, growing, and aging longer due to better medical care; people tend to sustain their SCI at a higher age, and the process of natural ageing likely interferes with the ambulation capacity and performance.

As clinical rehabilitation practice should be evidence-based there is also a demand for researchers not only to strive for originality and novelty but also to contribute to increasing the knowledge base and reproduce research. And, as we mentioned, the body of research in this population is still rather small compared to other more common patient populations.

In the revised manuscript, we have written much more clearly how this study can be expected to add to the research and clinical body of knowledge. Please see Expected Results and Significance sections.

Reviewer 2 Report

Comments and Suggestions for Authors

The study presents an in-depth examination of ambulatory performance, aiming to enhance participation, improve quality of life, and maximise everyday functioning, particularly for individuals with spinal cord injury (SCI).

While the application of machine learning to improve ambulatory performance in SCI individuals represents a step towards innovative research, this approach is not unprecedented. The document suggests that such analytics are part of an emerging trend rather than a pioneering endeavour. To enhance originality, the study could delve into untapped areas of machine learning or incorporate cutting-edge technology like neuromuscular electrical stimulation algorithms that haven't been widely explored in this context.

The methodology section, though comprehensive, lacks sufficient detail regarding the statistical validation of the machine learning algorithms used and their predictive accuracy. For a study of this nature, it is crucial to demonstrate the robustness of the methods through rigorous validation methods, cross-validation techniques, and an in-depth explanation of algorithm selection. This would not only enhance the scientific rigour but also the replicability of the research.

The potential impact of this study on enhancing the quality of life for individuals with SCI is clear and noteworthy. However, the significance is slightly overshadowed by a lack of direct application or discussion about how these machine learning insights translate into practical outcomes for patients. For a higher significance score, the study could extend its findings to recommend specific interventions, rehabilitation techniques, or the development of new assistive devices informed by the research data.

The study presents a commendable attempt to investigate ambulatory performance using machine learning techniques but falls short in areas that are critical for a rigorous, original, and highly significant contribution to the field. To achieve a state of minor revision, the authors should focus on enhancing the originality by exploring more groundbreaking approaches or techniques. In terms of rigour, a detailed validation of the machine learning models and a deeper dive into their predictive performance are needed. Lastly, to increase its significance, the research should make stronger connections between its findings and their practical applications, potentially guiding the development of new rehabilitative technologies or interventions.

For inspiration and to strengthen the study, the authors should reference:

  • For innovative machine learning applications: Machine learning in physical rehabilitation. This could provide insights into new or less-explored algorithms and approaches.
  • For methodological rigor and validation: Gait analysis techniques and technologies in SCI. This reference could offer a deeper understanding of how to rigorously validate machine learning models in the context of gait analysis.
  • For practical significance and applications: Improving health outcomes in SCI. This paper might offer examples of how to directly relate research findings to improvements in patient care and rehabilitation techniques.

Incorporating these insights could significantly bolster the manuscript, providing a clearer path toward minor revisions with an emphasis on enhancing the originality, rigour, and significance of the work.

Author Response

Thank you for thoughtful feedback. We have revised according to your comments and hope you agree that the manuscript has improved.

Comments and Suggestions for Authors

The study presents an in-depth examination of ambulatory performance, aiming to enhance participation, improve quality of life, and maximise everyday functioning, particularly for individuals with spinal cord injury (SCI).

While the application of machine learning to improve ambulatory performance in SCI individuals represents a step towards innovative research, this approach is not unprecedented. The document suggests that such analytics are part of an emerging trend rather than a pioneering endeavour. To enhance originality, the study could delve into untapped areas of machine learning or incorporate cutting-edge technology like neuromuscular electrical stimulation algorithms that haven't been widely explored in this context.

Response: Thanks for this interesting point. While machine learning methods have been around a while, their application in human movement studies in clinical treatment and research has certainly not been exhausted. The sheer magnitude of data also lends itself well to data-driven analyses, given that enough data is available.  In the revised manuscript, we have included a longer introduction and motivation to our planned approach and described the expected outcome in more detail. This study will not be an intervention study; sadly your second suggestion, a neuromuscular electrical stimulation intervention study, certainly would be exciting but is not in our immediate horizon.

The methodology section, though comprehensive, lacks sufficient detail regarding the statistical validation of the machine learning algorithms used and their predictive accuracy. For a study of this nature, it is crucial to demonstrate the robustness of the methods through rigorous validation methods, cross-validation techniques, and an in-depth explanation of algorithm selection. This would not only enhance the scientific rigour but also the replicability of the research.

Response: We have written much more detail about the planned studies with machine learning, including how to test their robustness, in section 3.

The potential impact of this study on enhancing the quality of life for individuals with SCI is clear and noteworthy. However, the significance is slightly overshadowed by a lack of direct application or discussion about how these machine learning insights translate into practical outcomes for patients. For a higher significance score, the study could extend its findings to recommend specific interventions, rehabilitation techniques, or the development of new assistive devices informed by the research data.

The study presents a commendable attempt to investigate ambulatory performance using machine learning techniques but falls short in areas that are critical for a rigorous, original, and highly significant contribution to the field. To achieve a state of minor revision, the authors should focus on enhancing the originality by exploring more groundbreaking approaches or techniques. In terms of rigour, a detailed validation of the machine learning models and a deeper dive into their predictive performance are needed. Lastly, to increase its significance, the research should make stronger connections between its findings and their practical applications, potentially guiding the development of new rehabilitative technologies or interventions.

Response: Thanks again for constructive feedback. To know exactly how to implement new knowledge from research is often challenging and takes time, and implementation is of course a research field of its own. We believe that our planned findings will be to treatment planning, targeted rehabilitation, and design of technical aids, for instance. In the revised manuscript, we have clarified this.

For inspiration and to strengthen the study, the authors should reference:

  • For innovative machine learning applications: Machine learning in physical rehabilitation. This could provide insights into new or less-explored algorithms and approaches.

  • For methodological rigor and validation: Gait analysis techniques and technologies in SCI. This reference could offer a deeper understanding of how to rigorously validate machine learning models in the context of gait analysis.

  • For practical significance and applications: Improving health outcomes in SCI. This paper might offer examples of how to directly relate research findings to improvements in patient care and rehabilitation techniques.

Incorporating these insights could significantly bolster the manuscript, providing a clearer path toward minor revisions with an emphasis on enhancing the originality, rigour, and significance of the work

Response: We have added around 20 new references to strengthen the manuscript. Not specifically these three, but we hope the reviewer will agree that they have strengthened the manuscript.

Round 2

Reviewer 2 Report

Comments and Suggestions for Authors

The document covers the incorporation of machine learning techniques to enhance clinical understanding and management of Spinal Cord Injury (SCI), which is a relatively innovative approach. The use of tree ensembles like random forests and XGBoost for classification and regression tasks highlights a novel integration of advanced machine learning models within the clinical research setting on SCI. The concept of exploring gait patterns and applying data-driven machine learning techniques for analysis, as mentioned, further emphasizes its originality.

The methodology entails a comprehensive combination of statistical methods (e.g., Spearman, multiple regressions) and machine learning algorithms (including k-means and hierarchical clustering). The use of cross-validation to benchmark models and parametric/non-parametric tests tailored to data normality shows a detailed and methodically sound approach. Assessing motion parameters and experimental validations like the use of Gait Deviation Index for examining gait patterns ensures a rigorous scientific approach. However, more detail on the exact processes of model validation and parameter tuning would enhance thesection.

The potential impact of this research is substantial. The application of these advanced techniques could lead to significant improvements in diagnosing and managing SCI, by defining more personalized treatment plans and rehabilitation strategies. The exploration of associations between gait performance and muscular or sensory impairments could provide key insights that are clinically valuable. This could substantially benefit the clinical community and enhance patient care for individuals with SCI.

 Overall, the document strides towards a significant advancement in understanding and managing spinal cord injuries through innovative applications of machine learning. This work could significantly influence patient outcomes by optimizing and personalizing rehabilitation strategies. However, to further solidify its position within the field, a more detailed exposition of model validation techniques and a deeper dive into data handling would be beneficial.

Comparing to known methods such as those discussed in the paper  published in Applied Sciences (https://www.mdpi.com/2076-3417/10/18/6168), this document can be cited and used to strenghten even more the paper since it uses a comparative innovative approach to machine learning. Yet, it enhances its contribution by specifically focusing on SCI and multi-faceted data integration (muscle strength, sensory and motor levels, etc.) which are crucial for biomechanical outcomes. The document significantly contributes to the personalized treatment approaches, which is a step beyond the general applications reviewed in the aforementioned paper. 

By expanding on biomechanical outcomes such as detailed gait analysis and the association of these outcomes with orthotic intervention possibilities, the paper not only adds depth to the clinical utility of gait analysis but also highlights potential areas for developing smart orthoses based on individual biomechanical profiles observed through machine learning-enhanced methodologies. This specific focus can significantly strengthen the impact of the research in clinical settings, making it a valuable asset for personalized medicine in SCI rehabilitation.  

Author Response

Thank you for the constructive feedback and suggested article.  We made 2 changes based on the comments:

Comment: "Comparing to known methods such as those discussed in the paper  published in Applied Sciences (https://www.mdpi.com/2076-3417/10/18/6168), this document can be cited and used to strenghten even more the paper since it uses a comparative innovative approach to machine learning."

Response: We have added a citation of this paper and the methods it describes to use robotics not only directly in rehabilitation but even to measure and assess physical properties that are otherwise difficult to measure in a patient population, but useful in these and similar methods. The new text is shown here in red:

Regaining and optimizing ambulatory function after SCI has been a major goal in both rehabilitation and research over the years, but as both the prevalence of persons with incomplete SCI [12] and age at injury [11] are increasing, the physical prerequisites for regaining and refining ambulatory function is also changing. There has been a focus on advanced equipment such as body weight-supported treadmill training, robotics and exoskeletons in recent research. In addition to their direct use in rehabilitation, robotics can also be useful in rehabilitation by assessing stiffness, joint torque, and range of motion in patient populations, metrics that are not trivial to assess; a recent report describes their use in a stroke population [16]. In spite of their advantages of providing an opportunity for massive and repetitive gait training, their high cost and need for staff with high technical skills [17] make their frequent use impractical on a large scale outside the rehabilitation centers. It thus remains important to perform continuous research regarding overground ambulation for persons who walk in daily life. Moreover, there are rapid developments in technology to measure and analyze movements, muscle function and forces during ambulation. We believe that leveraging the use of new technologies with the change in patient characteristics could bridge the gap of knowledge in this field.

Comments"However, more detail on the exact processes of model validation and parameter tuning would enhance the section."

and

"However, to further solidify its position within the field, a more detailed exposition of model validation techniques and a deeper dive into data handling would be beneficial."

Response: we have clarified how we plan to tune, optimize, and validate hyperparameters in the machine learning models.  Page 8: 

For classification and regression applications, literature has shown that tree-ensembles models are effective on tabular data [45]. Therefore, we plan to investigate several tree-ensembles models such as random forests [46] and XGBoost [47] for these applications and train them through our collected data sets. Several metrics will be used to evaluate model performance and select the best for each application. For classification, evaluation metrics such as accuracy score, precision score, and recall score will be considered. Accuracy reflects overall correct predictions while precision finds models minimizing Type I errors. Recall identifies models with fewest Type II errors. For regression, root mean square error can be used to quantify average errors between predicted and actual values. R-squared determines quality of fitness by variation explained. In both applications, we will perform grid searches for fine tuning hyperparameters while training. For example, key hyperparameters for random forests such as number of trees, maximum tree depth, and measures of split quality can be investigated. By exploring this hyperparameter space, optimal configuration for each model can be achieved. Cross-validation will then be used to compare algorithms across relevant metrics. The classifiers or regressors averaging the highest scores will be determined as the most appropriate models for each application.

Thanks again for your comments and we hope we have adequately addressed them.